# Assessing the characteristics of un- and under-vaccinated children in low- and middle-income countries: A multi-level cross-sectional study

C. Edson Utazi[1]*, Oliver Pannell[2], Justice M. K. Aheto[1], Adelle Wigley[1], Natalia Tejedor-Garavito[1], Josh Wunderlich[3], Brittany Hagedorn[4], Dan Hogan[3], Andrew J. Tatem[1]

**1** WorldPop, School of Geography and Environmental Science, University of Southampton, Southampton, United Kingdom, **2** Flowminder Foundation and WorldPop, School of Geography and Environmental Science, University of Southampton, Southampton, United Kingdom, **3** Gavi, The Vaccine Alliance, Geneva, Switzerland, **4** Institute for Disease Modeling, Bill & Melinda Gates Foundation, Seattle, Washington, WA, United States of America

\* c.e.utazi@soton.ac.uk

**Data Availability Statement:** The R code used in the analysis is available at: https://github.com/EdsonUtazi/NCOMMS_Multmod_paper_code. All

## Abstract

Achieving equity in vaccination coverage has been a critical priority within the global health community. Despite increased efforts recently, certain populations still have a high proportion of un- and under-vaccinated children in many low- and middle-income countries (LMICs). These populations are often assumed to reside in remote-rural areas, urban slums and conflict-affected areas. Here, we investigate the effects of these key community-level factors, alongside a wide range of other individual, household and community level factors, on vaccination coverage. Using geospatial datasets, including cross-sectional data from the most recent Demographic and Health Surveys conducted between 2008 and 2018 in nine LMICs, we fitted Bayesian multi-level binary logistic regression models to determine key community-level and other factors significantly associated with non- and under-vaccination. We analyzed the odds of receipt of the first doses of diphtheria-tetanus-pertussis (DTP1) vaccine and measles-containing vaccine (MCV1), and receipt of all three recommended DTP doses (DTP3) independently, in children aged 12–23 months. In bivariate analyses, we found that remoteness increased the odds of non- and under-vaccination in nearly all the study countries. We also found evidence that living in conflict and urban slum areas reduced the odds of vaccination, but not in most cases as expected. However, the odds of vaccination were more likely to be lower in urban slums than formal urban areas. Our multivariate analyses revealed that the key community variables—remoteness, conflict and urban slum—were sometimes associated with non- and under-vaccination, but they were not frequently predictors of these outcomes after controlling for other factors. Individual and household factors such as maternal utilization of health services, maternal education and ethnicity, were more common predictors of vaccination. Reaching the Immunisation Agenda 2030 target of reducing the number of zero-dose children by 50% by 2030 will require country tailored

the data used in this work are publicly available via the sources referenced in the Methods section. These can be obtained from the authors upon request.

**Funding:** This work was funded by the Bill & Melinda Gates Foundation (BMGF) and Gavi, the Vaccine Alliance (Investment ID: INV-002397) through a grant awarded to A.J.T, C.E.U and N.T.-G. The funders had no role in study design, data collection and analysis, decision to publish, or preparation of the manuscript.

**Competing interests:** I have read the journal's policy and the authors of this manuscript have the following competing interests: D.H. and J.W. work for Gavi, the Vaccine Alliance, while B.H. works for BMGF. The results and conclusions contained here are those of the authors and do not necessarily reflect the position or policies of Gavi, the Vaccine Alliance and BMGF. The authors declare no other competing interests.

analyses and strategies to identify and reach missed communities with reliable immunisation services.

## Introduction

Since the establishment of the Expanded Programme on Immunization in 1974 by the World Health Organization (WHO), there have been remarkable increases in immunization coverage globally, making childhood vaccination one of the most successful public health interventions [1, 2]. However, while there have been impressive increases in the coverage of newer vaccines like those protecting against pneumococcal pneumonia and rotavirus, increasing the coverage of routine immunisation (RI) to reach the last 20% of children in low- and middle-income countries (LMICs) has proved more challenging [2, 3]. In 2019, 19 million children were not fully vaccinated with all three recommended DTP doses, and of those, 72% (13.6 million) were zero-dose children who did not receive any dose of diphtheria-tetanus-pertussis (DTP) vaccine [4]. In 2020, these figures increased to 22.7 million children and 75% (17.1 million), respectively, due to the disruptions to health services caused by the ongoing coronavirus pandemic [4, 5]. Coverage varies across regions and countries, with lower coverage occurring mostly in LMICs, with five countries (Nigeria, India, Democratic Republic of Congo, Pakistan and Ethiopia) accounting for 50% of all zero-dose children globally in 2020 [4]. More recently, geospatial analysis of coverage in various LMICs has revealed significant geographic inequities in coverage achieved through both RI and vaccination campaigns [3, 6, 7]. The persistence of geographical inequities invariably undermines efforts towards the attainment of national coverage targets and the prevention of deaths and disease outbreaks within countries.

Inequities most often negatively affect populations characterized by poverty, overcrowding, poor sanitation practices, lack of access to basic healthcare services, civil/political unrest etc., with the consequential high risks of transmission of vaccine-preventable diseases and outbreaks [8]. Of interest in this work are marginalized populations living in remote-rural, urban slum and conflict-affected areas, as has been identified by the Equity Reference Group for Immunization (ERG) [9, 10]. A preceding study [11] found that out of 99 LMICs, 6% and 15% of DTP zero-dose children lived in conflict-affected areas when using the 'narrow' and 'broad' conflict definitions, respectively. Also, the study found that over 11.8%, 13.1% and 12.9% of children who had not received DTP1, DTP3 and MCV1, respectively, lived in remote rural areas that were more than 3 hours travel time from the nearest town or city, while more than 19% lived in urban areas and more than 7% in peri-urban areas. There is currently a high level of interest in the development of strategies and interventions that target these at-risk populations, stimulating the need for a current and robust evidence base regarding their population sizes, geographic distribution, immunisation levels, and other characteristics. Fast-tracking progress for these marginalized populations is vital to achieving the WHO's Immunisation Agenda 2030 target of a 50% reduction in zero-dose children by 2030, as well as targets within the Sustainable Development Goals [12] and Gavi, the Vaccine Alliance's 2021–2025 Strategy [13].

A plethora of studies, including systematic reviews, have established linkages between several contextual factors and indicators of vaccination coverage in different LMICs [14–17]. However, to date, there exists no in-depth quantitative study utilizing cross-sectional individual-level data with a focus on exploring key non- and under-vaccination characteristics simultaneously across the ERG's priority settings [10].

Here, we build on the current literature by developing a robust framework to investigate and quantify the relationships between non- and under-vaccination and key community variables–remoteness, conflict and urban slum—both when and when not controlling for other factors in nine LMICs. We determine predictors of non-vaccination (or zero dose) and under-vaccination by analyzing the following outcomes independently in children aged 12–23 months: receipt of the first dose of diphtheria-tetanus-pertussis (DTP1) vaccine, receipt of the first dose of measles-containing vaccine (MCV1), and receipt of all three recommended DTP doses (DTP3). The consideration of all three outcomes simultaneously enables us capture factors that determine non- and under-vaccination close to the beginning and towards the end of the childhood vaccination series [18].

## Methods

### Data

**Outcome variables and DHS covariates.** Cross-sectional data were obtained from the most recent Demographic and Health Surveys (DHS) conducted between 2008 and 2018 in the nine study countries, namely Nigeria (2018), Cambodia (2014), Pakistan (2017–18), Ethiopia (2016), Zambia (2018), India (2015–16), Democratic Republic of Congo (DRC, 2013–14), Mozambique (2011) and Madagascar (2008–09) [19]. These countries were selected due to (i) representing a wide range of low/middle income geographies (West, Southern and East Africa, South and Southeast Asia) as well as a wide range of coverage rates (Table 1), and (ii) being high priority countries for implementing/donor organizations such as Gavi and the Bill and Melinda Gates Foundation, and, therefore, also the focus of previous work [7, 20–22].

For each of the three vaccine doses–DTP1, DTP3 and MCV1, data on the vaccination status of each sampled child aged 12–23 months were extracted from the respective DHS databases. We considered evidence of vaccination obtained from a vaccination card as well as through caregiver recall. Each of these three outcome variables were categorized into two levels–not vaccinated (reference category) or vaccinated for each sampled child. We also extracted DHS data on a wide range of individual, household and community level covariates. The selection of these covariates was guided by data availability, evidence from the literature [15–17] and expert knowledge. The full range of explanatory variables considered and how these were coded in our analyses are provided in Fig 1 and Table A in S1 Text. Additional information on covariate selection and data processing is provided in S1 Text.

**Table 1. Summary of DHS data and direct (or survey-weighted) estimates of DTP1, 3 and MCV1 coverage for all nine study countries.**

| Country | Unweighted number of children aged 12–23 months | Number of complete cases | Number of DHS variables analysed | DTP1 coverage (%) | DTP3 coverage (%) | Dropout rate* between DTP1 and DTP3 (%) | MCV1 coverage (%) |
|---------|---|---|---|---|---|---|---|
| Cambodia | 1,441 | 1,377 | 21 | 94·0 | 83·7 | 11·0 | 78·6 |
| DRC | 3,182 | 2,948 | 24 | 81·2 | 60·5 | 25·5 | 71·6 |
| Ethiopia | 1,868 | 1,757 | 22 | 73·2 | 53·2 | 27·3 | 54·3 |
| India | 49,056 | 46,130 | 20 | 89·5 | 78·4 | 12·4 | 81·1 |
| Madagascar | 2,145 | 2,039 | 20 | 84·2 | 72·8 | 13·5 | 69·6 |
| Mozambique | 2,221 | 2,110 | 20 | 91·3 | 76·2 | 16·5 | 81·5 |
| Nigeria | 6,036 | 5,704 | 24 | 65·3 | 50·1 | 23·3 | 54·0 |
| Pakistan | 2,312 | 2,035 | 21 | 86·3 | 75·4 | 12·6 | 73·2 |
| Zambia | 1,897 | 1,818 | 23 | 97·9 | 92·1 | 5·9 | 90·9 |

*Relative to DTP1 coverage.

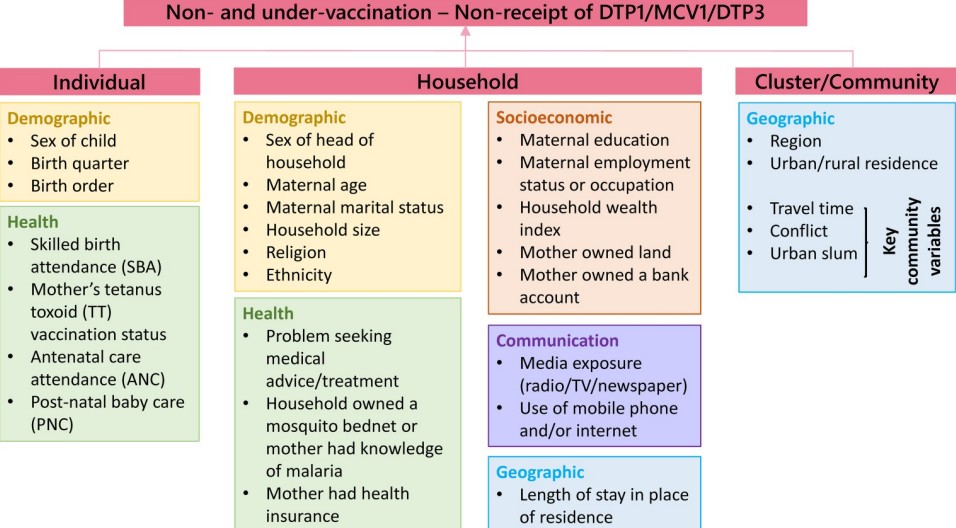

**Fig 1. Outcome variables and groups of individual, household and community level covariates considered in the study.** Further details are provided in Table A in S1 Text.

These additional processing steps and those outlined in the Analysis section resulted in different numbers of DHS covariates included in the analyses for each country, as reported in Table 1. The numbers of children with complete records (including the availability of the geographical coordinates of the survey cluster the child was sampled from) in the study ranged from 1,377 in Cambodia to 46,130 children in India. We note that due to small numbers of unvaccinated children, DTP1 was not included as an outcome variable in the analyses for Cambodia and Zambia.

The range of scenarios represented by the study countries is apparent in the coverage rates reported in Table 1. Zambia had the highest coverage levels and the lowest dropout (or under-vaccination) rate, with > 90% coverage for each of the three vaccine doses; while Nigeria and Ethiopia had the poorest coverage levels and were among the highest dropout rates. The lowest dropout rates (<13%) were observed in Cambodia, India and Pakistan. Also, no less than 80% coverage was recorded for at least two vaccines in Cambodia, India and Mozambique.

**Key community variables.** To explore potential areas where un- and under-vaccinated children may reside, we obtained geospatial covariate datasets on travel time to populated areas with at least 50,000 people (travel time—used as a proxy for remoteness [23]), conflict-affected areas [24] and urban slum areas [19]. Travel time data were acquired at 1 km resolution (Fig A in S1 Text), and following approaches outlined in Perez-Haydrich et al. [25], the data were processed for each country to extract the corresponding values for the DHS cluster locations. The extracted cluster/community level data were then grouped into three classes–lower, medium and higher travel times—using the terciles of their distributions (see Table B in S1 Text) for each country. Data on conflicts occurring within the 2 years prior to each DHS survey were obtained from the Armed Conflict Location and Event Data Project (ACLED) [24] and aggregated to the second administrative (admin 2) level within each country, using the global database of Global Administrative areas (GADM) (https://gadm.org/data.html). The 2-year period considered provides an opportunity to capture all conflict events that may have affected the 12–23 month cohort from birth. The aggregation to admin 2 areas was chosen as these constitute the geographic units most relevant for program planning and administration in many LMICs, so conflicts within them likely impact operations within the entire district.

Conflict-affected administrative areas were then classified using a rate-based approach as areas that had > 30 deaths per 1 million population due to conflict [11] (a narrower threshold of > 300 deaths per 1 million was used in a sensitivity analysis–see analysis results). The population data used for this classification were obtained from WorldPop [26]. Subsequently, each DHS cluster location was classified as a conflict area if it fell within a conflict administrative area and as a non-conflict area otherwise.

Further, for each country, the DHS cluster locations were classified as urban slum or non-slum areas using the most widely adopted UN-Habitat definition [27, 28] of slum dwellings and approaches outlined elsewhere [29], to enable a consistent classification of these deprived populations across the countries–see S1 Text for details.

All community level geospatial data–original and processed data—are mapped in Figs A-D in S1 Text. We note that based on these definitions, no cluster location was classified as a conflict area in Cambodia and Zambia, and urban slum was not considered in the analysis for Cambodia due to sample size limitation (see also Table D in S1 Text). To further facilitate comparisons of the odds of vaccination between urban slums and formal urban areas (and between formal urban and rural areas), we created a categorical 'place of residence' variable with three levels: formal urban, urban slum and rural areas, setting formal urban area as the reference category. This variable was considered in bivariate analyses only (see statistical analysis section).

## Statistical analysis

**Descriptive statistics, bivariate analysis and other exploratory analyses.** For each country-vaccine combination, the processed data were summarized using frequencies and percentages. We fitted frequentist simple binary logistic regression models with each of the outcomes to explore their relationships with the covariates. These models are reduced versions of the full multivariate model (see eq (1) in S1 Text) involving only the first level (i.e. the individual level) and a single covariate at a time, and are useful for understanding how each covariate relates to the outcome variable without the interference of other covariates. We also tested for (multi) collinearity in the covariates as discussed in S1 Text.

**Multivariate analysis using multi-level modelling.** To estimate the relationships between the outcome variables and the covariates in a multivariate setting, we adopted a Bayesian multi-level random intercept logistic regression model, accounting for individual, household, community and stratum level variation. A detailed description of the model and model estimation is provided in S1 Text. Also, in S1 Text, we discuss the metrics used to: evaluate the contribution of the key community variables (remoteness, conflict and urban slum) to explaining the variation in the outcome variables, assess the proportion of the total residual variation lying at the various levels of the model's hierarchy, and evaluate the predictive ability of the fitted models.

In both the bivariate and multivariate analyses, we calculated the crude/unadjusted and adjusted odds ratios (cORs and aORs) as the exponents of the estimates of the fixed effects and their corresponding 95% confidence or credible intervals (CIs) to evaluate the significance of the associations between the covariates and non- and under-vaccination. In both bivariate and multivariate analyses, covariates whose 95% CIs did not include one had a significant association with vaccination.

## Results

### Fixed effects—Crude and adjusted odds ratios and associated uncertainties

In bivariate analyses, Tables E-M in S1 Text show that for Nigeria and India, all the individual and household level factors, except sex of child, were significantly associated with the receipt

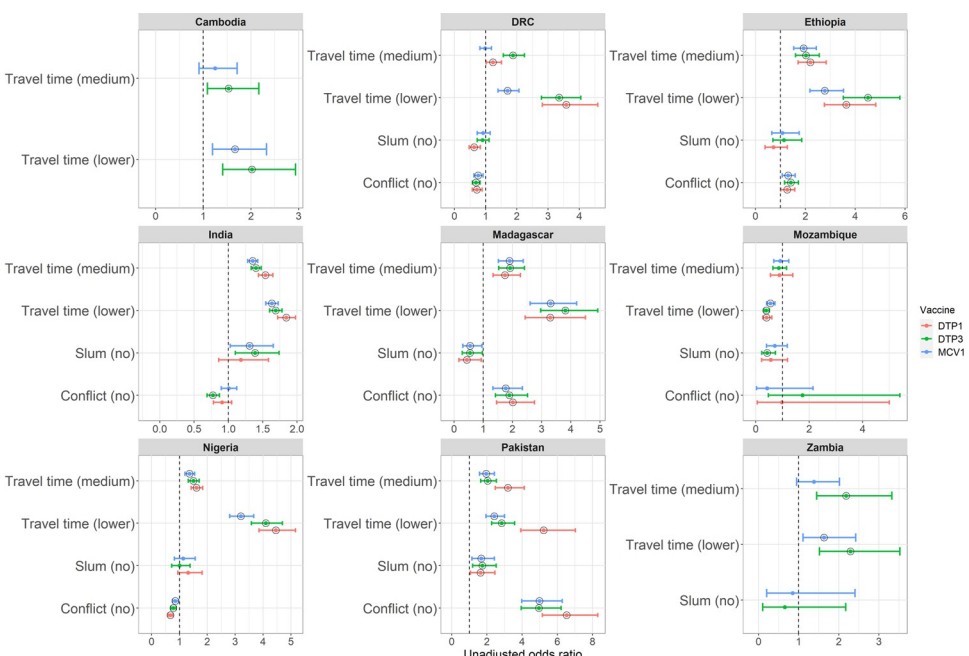

**Fig 2. Plots of crude/unadjusted odds ratios and corresponding 95% CIs.** The plots are based on bivariate analyses between each of the community-level variables (remoteness measured using travel time to the nearest city, urban slum and conflict area) and receipt of DTP1, DTP3 and MCV1 vaccinations. The vertical dotted lines mark the odds ratio of 1. The black circles show significant associations. Note that the reference category for travel time is 'higher'.

of at least one of the three vaccine-dose combinations, justifying their consideration in our work. For other countries with smaller sample sizes (as given in Table 1), there were more individual and household level factors without a significant association with vaccination, notable among which are sex of child, maternal marital status, sex of head of household and maternal employment status. These covariates were not significantly associated with any of the outcome variables in four of the study countries.

The associations (unadjusted ORs and 95% CIs) between the key community variables and the odds of vaccination in bivariate analyses are additionally displayed in Fig 2. Remoteness increased the odds of non- and under-vaccination in all the countries except Mozambique. Living in a conflict area significantly reduced the odds of vaccination in Ethiopia, Madagascar, Pakistan and India (DTP3 only). Conflict also had significant associations with vaccination in DRC and Nigeria but there was no evidence of reduction in the odds of vaccination. However, when using the narrower threshold to define conflict-affected areas (Tables D and O in S1 Text), we found that conflict significantly reduced the odds of vaccination in Nigeria but the results remained unchanged for DRC and other countries. Living in a slum area, compared to a non-slum area, was significantly associated with decreased odds of vaccination in Pakistan and India (DTP3 only). It was also statistically significant in DRC (DTP1 only), Madagascar, and Mozambique (DTP3 only). Further, living in a slum area, compared to a formal urban area, led to significant decreases in the odds of vaccination in all cases except Madagascar, Mozambique (DTP1 and DTP3 only) and Zambia (see Table N in S1 Text). The odds of vaccination were also significantly lower in rural areas compared to both urban areas and formal urban areas (Tables E–N in S1 Text) for all country-vaccine combinations except Zambia (MCV1).

The aORs and corresponding 95% CIs from the multivariate analyses results are plotted in Fig 3 for Madagascar and Figs H–O in S1 Text for other country-vaccine combinations. For Madagascar, Fig 3 shows that remoteness was the only key community variable that had a significant association with non- and under-vaccination. Children living in areas with lower and medium travel times, compared to areas with higher travel times, had 128% (aOR = 2.28, 95% CI: 1.49–3.52) and 89% (aOR = 1.89, 95% CI: 1.32–2.71) higher chance of receiving MCV1, respectively. For DTP1, the odds of vaccination were 188% (aOR = 2.88, 95% CI: 1.60–5.30) and 96% (aOR = 1.96, 95% CI: 1.22–3.18) higher; and for DTP3, 232% (aOR = 3.32, 95% CI: 1.95–5.75) and 105% (aOR = 2.05, 95% CI: 1.32–3.20) higher, respectively. Other variables that had significant positive associations with MCV1, DTP1 and DTP3 vaccinations (i.e. significantly reduced the odds of non- and under-vaccination) were: skilled birth attendance, antenatal care attendance, postnatal care, birth quarter (July–September), maternal receipt of TT vaccination before birth, maternal education, religion (Christian) and maternal employment. Further, children from Atsinanana/Analanjirofo/Alaotra-Mangoro regions (see Fig E in S1 Text) had significantly higher MCV1 coverage. The odds of vaccination were significantly lower among children from smaller households (< = 4 members) for all three outcomes and for individual vaccines as follows: for MCV1—children from medium-sized households (5–8 members) and children with a birth order of at least 5; for DTP3—children from Atsimo-Andrefana/Androy/Anosy/Menabe regions; and for DTP1—children born to mothers who were divorced, widowed or had other marital status. Detailed interpretations of the estimated relationships for all country-vaccine combinations are provided in S1 Text and summarized in Fig 4A. Note that the reference categories of the covariates are already provided in Tables E–M in S1 Text. We now focus here on interpreting the broader patterns seen in these results.

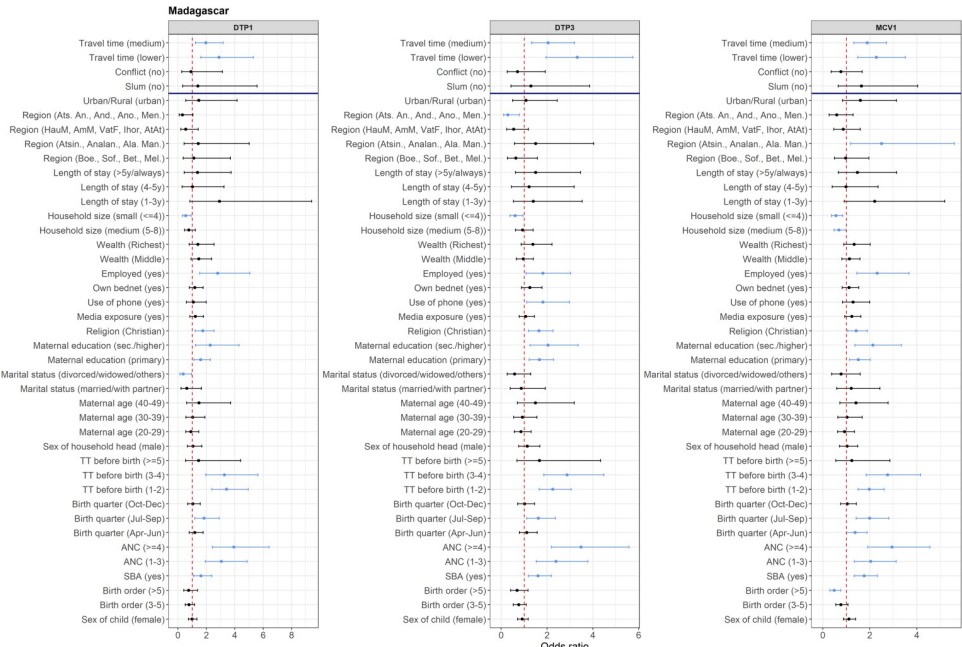

**Fig 3. Adjusted odds ratio (aOR) and corresponding 95% credible interval (95% CI) plots for Madagascar.** The vertical dotted red lines mark the odds ratio of 1. Light blue dots and lines show the aORs and 95CIs of variables that have significant associations with vaccination. A dark blue horizontal line separates the key community variables from other covariates. See S1 Text for the reference categories of the variables.

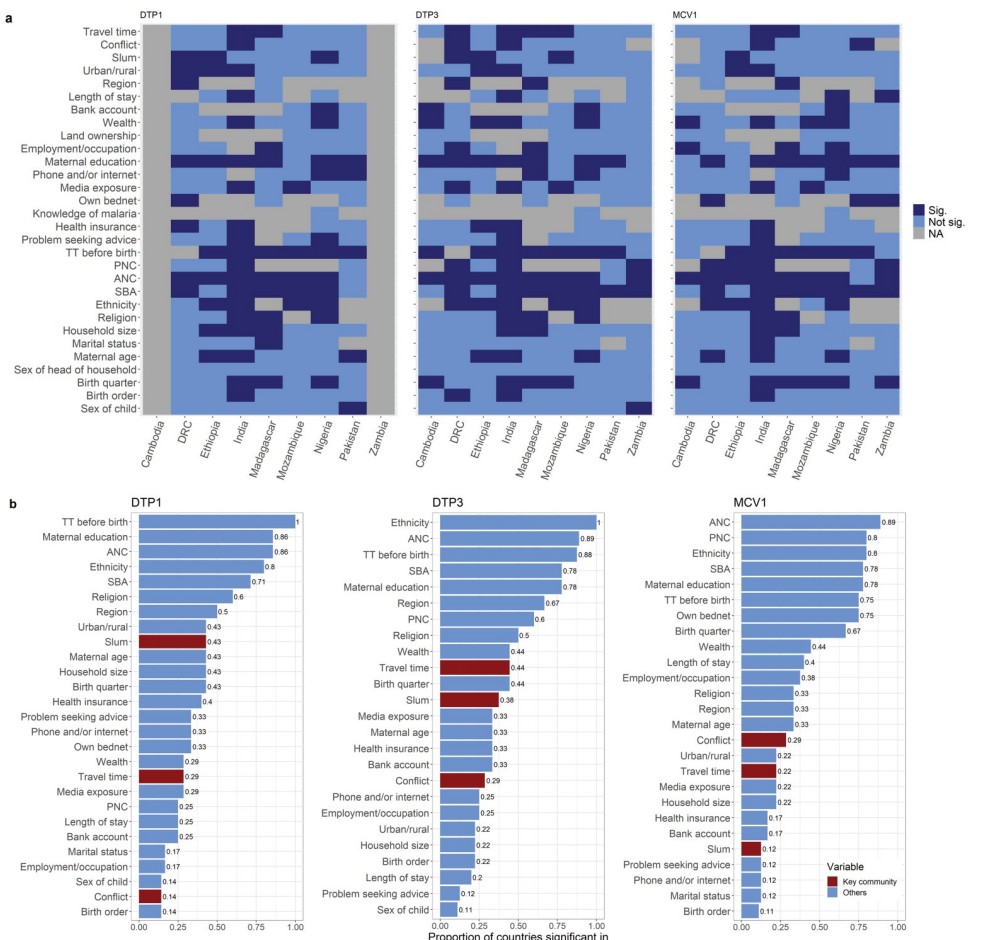

**Fig 4. Summaries of multivariate analyses results.** (a) A summary of the estimated relationships showing significant (sig.) and non-significant (not sig.) predictors of non- and under-vaccination in the multivariate analyses (missing variables are due to unavailability of data or multicollinearity); (b) Ranking of significant predictors of non- and under-vaccination in the study countries. For each vaccine dose, the ranks are based on the proportion of countries in which each variable was statistically significant in the multivariate analyses. Only variables found to be significant are shown.

In general, in multivariate analyses, the key community variables were significantly associated with non- and under-vaccination in DRC (travel time and conflict–DTP3, urban slum—DTP1 and DTP3), Ethiopia (urban slum–DTP1, DTP3 and MCV1), India (travel time and conflict–DTP1, DTP3, MCV1), Madagascar (travel time–DTP1, DTP3 and MCV1), Mozambique (travel time and urban slum–DTP3), Nigeria (urban slum–DTP1) and Pakistan (conflict–MCV1). However, remoteness significantly increased the odds of non- and under-vaccination in DRC, India and Madagascar only. Living in an urban slum significantly increased the odds of these events in DRC, Ethiopia and Nigeria only. Conflict significantly increased the odds of these events only in Pakistan, although conflict was not included in the analyses for Cambodia and Zambia as mentioned previously. Throughout, we observed only one instance with the key community variables (DRC—urban slum—DTP1) where the variable was a significant predictor of vaccination in both the bivariate and multivariate analyses but the direction of the relationship changed between both analyses. This may have been due to undetected collinearity (see S1 Text) or the effect of suppressor variables [30]. This rationale

also applies to other instances where this phenomenon was observed (e.g., the urban-rural covariate in India).

In Fig 4B, the covariates are also ranked based on the proportions of countries in which they were statistically significant predictors of non- and under-vaccination in multivariate analyses (the denominators were the numbers of countries in which the covariates were included in the analyses). The figure shows that compared to other factors, the key community variables were less frequent predictors of non- and under-vaccination in the study countries, with proportions less than 50%. Similarly, the urban-rural covariate was only significant in at most 43% of the study countries.

The topmost significant predictors of non- and under-vaccination were factors related to maternal utilization of health services (skilled birth attendance, antenatal care attendance, maternal receipt of tetanus toxoid vaccination and postnatal care), maternal education and ethnicity, all of which were significant in more than 50% of the study countries for all three outcomes, except postnatal care which had a much lower frequency for DTP1. In addition, religion and region of residence were significantly associated with the receipt of DTP1 and DTP3 in at least 50% of the countries in which these factors were included in the analyses, while ownership of a mosquito bednet and birth quarter had significant associations with MCV1 in more than 60% of the relevant study countries. In general, these high-ranking factors had positive associations with vaccination (i.e. the worse categories of these covariates increased the odds of non- and under-vaccination), except ethnicity and region whose categories were nominal.

Residual analysis and model evaluation results are reported and discussed in S1 Text.

## Discussion

Addressing recent policy directions within the global immunisation community, this study has evaluated the effects of key community variables (remoteness, conflict and urban slum) and a range of other factors on non- and under-vaccination in nine LMICs. These analyses are complimentary to global scale work which estimated the proportions and sizes of populations falling within these key risk groups [11].

In bivariate analyses, we found that remoteness increased the odds of non- and under-vaccination in all nine countries except Mozambique, which is potentially as a result of the fairly spatially homogeneous [7, 20] levels of vaccination coverage in this country. Further, we found evidence that living in conflict-affected and urban slum areas reduced the odds of vaccination, but not in most cases as expected. However, when comparing urban slums to formal urban areas, the odds of vaccination were consistently lower in slum areas in five out of the nine countries studied, three of which did not have evidence of lower odds of vaccination in slums compared to non-slum areas (i.e. rural and urban formal areas combined). Also, the odds of non- and under-vaccination were generally lower in rural areas compared to both urban and formal urban areas. The marginally lower odds of non- and under-vaccination estimated in conflict-affected areas in Nigeria and DRC and slum areas in Madagascar is likely due to: (i) under-sampling of areas with high intensity of conflicts in DHS surveys (see Figs B and D in S1 Text and DHS reports, e.g. [31]); (ii) an artefact of the conflict definition used in our analysis for Nigeria (a sensitivity analysis using a 'narrow' definition revealed a significant negative effect of conflict); and (iii) poor representation of urban slums in Madagascar (Madagascar had more than 77% rural population [32]).

When controlling for other factors, our multivariate analyses revealed that the key community variables–remoteness, conflict and urban slum–were sometimes associated with non- and under-vaccination, but they were not frequently predictors of these outcomes as anticipated.

Also, despite significant urban-rural differences in the odds of vaccination observed in all the countries in bivariate analyses, the urban-rural covariate also had reduced frequency of significance in multivariate analyses. Further, in multivariate analyses, we found that the predictors of non- and under-vaccination were broadly similar across the study countries, irrespective of whether these were high- or low-performing countries, but there were also some differences. Several gender-related factors were common predictors of non- and under-vaccination, including maternal utilization of health services (skilled birth attendance, antenatal care attendance, maternal receipt of tetanus toxoid vaccination, and postnatal care), maternal education and ethnicity. These factors were more likely to be significant across countries than the key community variables (except postnatal care, which had a lower frequency of significance for DTP1 than urban slum and remoteness), similar to findings reported in previous studies [14, 15]. Religion and region of residence were additionally frequent predictors of receipt of the routine doses–DTP1 and DTP3—given close to beginning of the vaccination series, but not MCV1 which is administered towards the end of the vaccination series both via RI and campaigns. This potentially highlights the role that campaigns can play in improving equity in vaccination coverage [7] and the need for improvements in RI services among underserved population groups [33]. Other frequent predicators of MCV1 vaccination were ownership of a mosquito bednet and birth quarter. The latter demonstrates the seasonality of MCV1 vaccination, which has previously been linked to agricultural activities, rainy season, etc [34]. Insecticide-treated bednets are often deployed via similar delivery mechanisms as MCV [35, 36], which likely explains its association with MCV.

By evaluating the significance of the key community variables and reinforcing the multiplicity of factors responsible for non- and under-vaccination in LMICs, our study points to the importance of country-tailored subnational assessments of immunisation coverage and interventions to address the barriers to immunisation. The key community variables are valuable, as has also been demonstrated in an earlier work [11] which found considerable proportions of zero-dose children (>11% in remote-rural areas) living in the different at-risk areas, but due consideration should be given to other factors. Notably, several factors related to gender emerged as important predictors of zero-dose and under-vaccinated children, and given the clear associations between receipt of vaccination and nearly all other indicators of other health services, integrated delivery approaches, either through fixed site or outreach strategies, merit more consideration. Also, the development of a composite zero-dose vulnerability index that enables the integration of all major risk factors within each country can be a useful tool for prioritizing subnational areas for interventions.

Our analyses were undertaken using data from multiple sources which are subject to certain limitations that should be taken into account when evaluating the findings. The sampling frames used for the various DHS surveys may have missed or underrepresented important at-risk groups living in urban slums and conflict-affected areas. We characterized remoteness based on travel time to the nearest city, but this could also be defined using travel time to the nearest health facility, although the latter may present considerable data quality issues [37]. The definition of a conflict-affected area is challenging. The data available [24] are not necessarily complete as they largely rely on aggregating media reports, and the areas within a country with conflict or insecurity challenges can change rapidly over time. The rate-based conflict definition used here was based on a given threshold that we considered reasonable. We considered a stricter threshold that would highlight only areas with the most intense conflict that is most likely to disrupt health services, but this would have led to substantial sample size limitations with respect to survey clusters falling in areas labelled as conflict-affected. These issues may have led to the poor identification of key at-risk populations in some of the countries.

Also, we were unable to model inter-country variation directly as our analyses were conducted for each individual country using approaches standardized across all the study countries.

Conclusively, our findings offer significant insights that can aid policy makers developing global and country-specific strategies to reach zero-dose chlidren and missed communities with sustainable RI services to improve equitable coverage. While understanding the distribution of missed children across remote-rural, urban slum and conflict areas can be helpful in some contexts, additional factors should be considered, tailored to a country's situation, to identify missed communities and design effective strategies to reach them.

## Supporting information

**S1 Text.** Supplementary file containing Tables A–O, Figs A-O and additional text referenced in the manuscript.
(DOCX)

## Author Contributions

**Conceptualization:** C. Edson Utazi, Dan Hogan, Andrew J. Tatem.

**Data curation:** C. Edson Utazi, Oliver Pannell, Justice M. K. Aheto, Adelle Wigley, Natalia Tejedor-Garavito, Josh Wunderlich.

**Formal analysis:** C. Edson Utazi, Justice M. K. Aheto, Dan Hogan.

**Methodology:** C. Edson Utazi.

**Visualization:** Oliver Pannell.

**Writing – original draft:** C. Edson Utazi, Dan Hogan, Andrew J. Tatem.

**Writing – review & editing:** C. Edson Utazi, Oliver Pannell, Justice M. K. Aheto, Adelle Wigley, Natalia Tejedor-Garavito, Josh Wunderlich, Brittany Hagedorn, Dan Hogan, Andrew J. Tatem.

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
