## [Decision Letter · Decision Letter 0]

26 Nov 2021

PGPH-D-21-00618

Assessing the characteristics of un- and under-vaccinated children in low- and middle-income countries: A multi-level cross-sectional study

Dear Dr. Utazi,

Thank you for submitting your manuscript to PLOS Global Public Health. After careful consideration, we feel that it has merit but does not fully meet PLOS Global Public Health’s publication criteria as it currently stands. Therefore, we invite you to submit a revised version of the manuscript that addresses the points raised during the review process.

It has been reviewed by experts in the field and we request that you make major revisions before it is processed further.

We look forward to receiving your revised manuscript.

Kind regards,

Eunha Shim

Academic Editor

Journal Requirements:

1. Please amend your detailed Financial Disclosure statement. This is published with the article, therefore should be completed in full sentences and contain the exact wording you wish to be published.

i). State the initials, alongside each funding source, of each author to receive each grant.

ii). State what role the funders took in the study. If the funders had no role in your study, please state: “The funders had no role in study design, data collection and analysis, decision to publish, or preparation of the manuscript.”

Reviewers' comments:

Reviewer's Responses to Questions

**Comments to the Author**

1. Does this manuscript meet PLOS Global Public Health’s publication criteria? Is the manuscript technically sound, and do the data support the conclusions? The manuscript must describe methodologically and ethically rigorous research with conclusions that are appropriately drawn based on the data presented.

Reviewer #1: Yes

Reviewer #2: Yes

Reviewer #3: Yes

2. Has the statistical analysis been performed appropriately and rigorously?

Reviewer #1: Yes

Reviewer #2: No

Reviewer #3: Yes

3. Have the authors made all data underlying the findings in their manuscript fully available (please refer to the Data Availability Statement at the start of the manuscript PDF file)?

Reviewer #1: Yes

Reviewer #2: Yes

Reviewer #3: Yes

4. Is the manuscript presented in an intelligible fashion and written in standard English?

Reviewer #1: Yes

Reviewer #2: Yes

Reviewer #3: Yes

5. Review Comments to the Author

Reviewer #1: Review for PGPH-D-21-00618 - Assessing the characteristics of un- and under-vaccinated children in low- and middle-income countries: A multi-level cross-sectional study

I appreciate the invitation to review this manuscript that addresses an important public health issue: vaccination coverage, especially in vulnerable populations living in low- and middle-income countries. The manuscript is original and has been well-designed methodologically.

I only have some general comments and some minor ones, as described below:

General comments

Descriptive studies have several important roles in medical and public health research. Typically, descriptive data provide information to develop hypotheses on causes or risk factors for illness (outcomes). In these sense, I strongly suggest that authors avoid using the term "risk" when describing and interpreting study results, since the risk can only be assessed in analytic studies. The authors must remember that, even using the odds ratio as a measure of association, this is a descriptive study.

This recommendation is directly in line with the objective of the study, which is well-proposed: "to investigate and quantify the relationships".

ABSTRACT

Findings: odds instead risk.

INTRODUCTION

Page 5, line 111: this is the first appearance of the acronym DRC, therefore, I suggest presenting the country name in full: Democratic Republic of the Congo [DRC].

METHODS

Page 5, lines 190-196: this paragraph should be moved to the Results.

DISCUSSION

Page 11, lines 369: odds instead risk.

Reviewer #2: This is a well-written paper about vaccinations in low and middle-income countries.

The authors claim that «Children living in remote-rural, urban slums and conflict- affected settings are more likely to be un- or under-vaccinated in some countries. Certain household characteristics, including lack of access to maternal health services and other gender-related barriers to vaccination, were consistent predictors of children missing out on vaccination, and our results suggest integrated delivery strategies deserve more consideration».

The results are done in several countries and are like what is known before. However, the title and the methods suggest that they did a multilevel study, and that they also used Bayesian multi-level binary logistic regression models. That could make this paper interesting and provide novel information.

Unfortunately, neither their methods nor the results describe the Bayesian multi-level binary logistic regressions. It seems as if a simple logistic regression has been done. Do the different inter and intra-country levels give important additional information about how vaccines are used?

Furthermore, it is also a requirement that when researchers use Bayesian statistics that they discuss the issue of prior probability. That has not been done in this study. Equally important, the results are not presented in a way that Bayesian statistics is usually done with credible intervals, and there is no information about the ICCs for the many levels that they studied.

Reviewer #3: The Paper focuses on characterising zero-dose and under vaccinated communities within LMIC countries using a appropriate methodology. The data used in in the public domain and the data and results support the conclusions made.

6. PLOS authors have the option to publish the peer review history of their article (what does this mean?). If published, this will include your full peer review and any attached files.

**Do you want your identity to be public for this peer review?** For information about this choice, including consent withdrawal, please see our Privacy Policy.

Reviewer #1: No

Reviewer #2: **Yes: **Bernt Lindtjorn

Reviewer #3: No

---

## [Decision Letter · Decision Letter 1]

20 Jan 2022

PGPH-D-21-00618R1

Assessing the characteristics of un- and under-vaccinated children in low- and middle-income countries: A multi-level cross-sectional study

Dear Dr. Utazi,

Thank you for submitting your manuscript to PLOS Global Public Health. After careful consideration, we feel that it has merit but does not fully meet PLOS Global Public Health’s publication criteria as it currently stands. Therefore, we invite you to submit a revised version of the manuscript that addresses the points raised during the review process.

We look forward to receiving your revised manuscript.

Kind regards,

Eunha Shim

Academic Editor

Journal Requirements:

Reviewers' comments:

Reviewer's Responses to Questions

**Comments to the Author**

1. If the authors have adequately addressed your comments raised in a previous round of review and you feel that this manuscript is now acceptable for publication, you may indicate that here to bypass the “Comments to the Author” section, enter your conflict of interest statement in the “Confidential to Editor” section, and submit your "Accept" recommendation.

Reviewer #1: All comments have been addressed

2. Does this manuscript meet PLOS Global Public Health’s publication criteria? Is the manuscript technically sound, and do the data support the conclusions? The manuscript must describe methodologically and ethically rigorous research with conclusions that are appropriately drawn based on the data presented.

Reviewer #1: Yes

3. Has the statistical analysis been performed appropriately and rigorously?

Reviewer #1: Yes

4. Have the authors made all data underlying the findings in their manuscript fully available (please refer to the Data Availability Statement at the start of the manuscript PDF file)?

Reviewer #1: Yes

5. Is the manuscript presented in an intelligible fashion and written in standard English?

Reviewer #1: Yes

6. Review Comments to the Author

Reviewer #1: Review for PGPH-D-21-00618-R1 - Assessing the characteristics of un- and under-vaccinated children in low- and middle-income countries: A multi-level cross-sectional study

The authors accepted most of the recommendations made in the first review, but some issues should be reviewed and discussed. Please see my comments in the attached file.

7. PLOS authors have the option to publish the peer review history of their article (what does this mean?). If published, this will include your full peer review and any attached files.

**Do you want your identity to be public for this peer review?** For information about this choice, including consent withdrawal, please see our Privacy Policy.

Reviewer #1: **Yes: **Everton Falcão de Oliveira

---

## [Editor Report · Decision Letter 2]

3 Feb 2022

Assessing the characteristics of un- and under-vaccinated children in low- and middle-income countries: A multi-level cross-sectional study

PGPH-D-21-00618R2

Dear Dr Utazi,

We are pleased to inform you that your manuscript 'Assessing the characteristics of un- and under-vaccinated children in low- and middle-income countries: A multi-level cross-sectional study' has been provisionally accepted for publication in PLOS Global Public Health.

Best regards,

Eunha Shim

Academic Editor